# IMPROVED COMMUNICATION LOWER BOUNDS FOR DISTRIBUTED OPTIMISATION

## ABSTRACT

Motivated by the interest in communication-efficient methods for distributed machine learning, we consider the communication complexity of minimising a sum of $d$-dimensional functions $\sum_{i=1}^{N} f_i(x)$, where each function $f_i$ is held by one of the $N$ different machines. Such tasks arise naturally in large-scale optimisation, where a standard solution is to apply variants of (stochastic) gradient descent. As our main result, we show that $\Omega(Nd \log d/\varepsilon)$ bits in total need to be communicated between the machines to find an additive $\epsilon$-approximation to the minimum of $\sum_{i=1}^{N} f_i(x)$. The result holds for deterministic algorithms, and randomised algorithms under some restrictions on the parameter values. Importantly, our lower bounds require no assumptions on the structure of the algorithm, and are matched within constant factors for strongly convex objectives by a new variant of quantised gradient descent. The lower bounds are obtained by bringing over tools from communication complexity to distributed optimisation, an approach we hope will find further use in future.

## 1 INTRODUCTION

The ability to distribute the processing of large-scale data across several computing nodes has been one of the main technical enablers of recent progress in machine learning, and the last decade has seen significant research effort dedicated to efficient distributed optimisation. One specific area of interest is *communication reduction* for distributed machine learning. Recently, several algorithms have been proposed to reduce the communication footprint of popular methods in machine learning and optimisation, in particular gradient descent and stochastic gradient descent; see e.g. Arjevani & Shamir (2015); Alistarh et al. (2017); Suresh et al. (2017); Tang et al. (2019) for recent work, and Ben-Nun & Hoefler (2019) for a survey. Despite this extensive work, less is known about theoretical limits of communication complexity of optimisation, especially in terms of *lower bounds* on the minimal number of bits which machines need to transmit to jointly solve an optimisation problem.

In this paper, we study this question in a classical distributed optimisation setting, where data is split among $N$ which that can communicate by sending point-to-point messages to each other. Given input dimension $d$, and a domain $\mathbb{D} \subseteq \mathbb{R}^d$, each machine $i$ is given an input function $f_i \colon \mathbb{D} \to \mathbb{R}$, and the machines need to jointly minimise the sum $\sum_{i=1}^{N} f_i(x)$, e.g. the empirical risk, with either deterministic or probabilistic guarantees on the output. The setting is a standard way to model the distributed training of machine learning models. For instance, if the individual loss functions are assumed to be (strongly) convex, we can model a classic regression setting, whereas if the function is non-convex, we can model distributed training of deep neural networks.

In this context, the key question is: what is the minimal number of bits which need to be exchanged for this optimisation procedure to be successful, and how does this number depend on the properties of the functions $f_i$, and the parameters $N$ and $d$?

### 1.1 OUR RESULTS

**Setting.** We consider this question in the classic message-passing model, where $N$ nodes communicate by sending messages to each other; specifically, each message is sent to a single receiver and not seen by the other nodes. Our complexity measure is the *total number of bits* sent by all the nodes.

Given this complexity measure, the model is equivalent (up to a constant factor) to a model where all messages are relayed via a special coordinator node, known in communication complexity as the *coordinator model* and in machine learning as the *parameter server model* (Li et al., 2014).

For convenience of presentation, we set $\mathbb{D} = [0, 1]^d$, and consider a problem where each node $i$ is given an input function $f_i\colon [0, 1]^d \to \mathbb{R}$, and the task is to approximate the minimum of the sum of the functions. That is, the coordinator needs to output $z \in [0, 1]^d$ and an estimate $r \in \mathbb{R}$ for the minimum function value such that

$$\sum_{i=1}^{N} f_i(z) \leq \inf_{x \in [0,1]^d} \sum_{i=1}^{N} f_i(x) + \varepsilon \quad \text{and} \quad \sum_{i=1}^{N} f_i(z) \leq r \leq \sum_{i=1}^{N} f_i(z) + \varepsilon. \tag{1}$$

Specifically, this models a standard distributed machine learning setting where we require one of the nodes to return the optimised final model, as well as the final value of the loss function. When proving lower bounds, we allow the nodes to compute arbitrary values that depend on the input functions and operate on real numbers; only the amount of communicated bits is limited. The precise definition is somewhat subtle, so we defer the details of the model to Section 2.

**Lower bounds for convex functions.** We show that, even if the input functions $f_i$ at the nodes are promised to be quadratic functions $x \mapsto \beta_0 \|x - x^*\|_2^2$ for a constant $\beta_0 > 0$, finding a solution satisfying (1) *deterministically* requires

$$\Omega\Big(Nd\log\frac{\beta d}{\varepsilon}\Big) \text{ total bits to be communicated,}$$

where $\beta = \beta_0 N$ is the smoothness parameter of $\sum_{i=1}^{N} f_i$, for parameters satisfying[1] $\beta d/\varepsilon = \Omega(1)$. For randomised algorithms, we give a lower bound of

$$\Omega\Big(Nd\log\frac{\beta d}{N\varepsilon}\Big) \text{ total bits to be communicated,}$$

for parameters satisfying $\beta d/N^2\varepsilon = \Omega(1)$. While this lower bound is slightly weaker due to the additional dependence in $N$, in most practical settings the number of parameters $d$ will be significantly larger than the number of machines $N$, multiplied by the error tolerance $\varepsilon$. (Specifically, in most practical settings $N \ll 1000$, whereas $\varepsilon \leq 10^{-3}$. More generally, it is sufficient that $d = \Omega(N^{2+\delta})$ for constant $\delta > 0$, for the randomised lower bound to match the deterministic one asymptotically.)

At a very high level, our results generalise the Tsitsiklis & Luo (1987) idea of linking the communication complexity with the number of quadratic functions with distinct minima in the domain. To extend this approach to the multi-node case $N > 2$ and to *randomised (stochastic)* algorithms, we build connections to results and techniques from *communication complexity*. Such connections have not to our knowledge been explored in the context of (real-valued) optimisation tasks, despite reductions from communication complexity being a standard lower bound technique e.g. in *distributed computing* (Das Sarma et al., 2012; Abboud et al., 2016; Drucker et al., 2014). Our work thus provides a model and a basic toolkit for applying communication complexity results to distributed optimisation, which should also be useful for understanding other optimisation tasks.

**Extensions.** While, for convenience, we work with functions over $[0, 1]^d$, our bounds immediately extend to arbitrary convex domains, as long as we can bound the number of functions with distinct minima in the domain. Beyond strongly convex and smooth functions, we also show that for non-convex $\lambda$-Lipschitz functions, solving (1) requires

$$N \exp\Big(\Omega\Big(d\log\frac{\lambda d}{\varepsilon}\Big)\Big) \text{ total bits communicated.}$$

The main takeaway from this result is that for non-convex objectives, one can induce exponentially higher communication cost by building convoluted input families where the coordinator is required to essentially learn all local input functions of the nodes.

---

[1]The constant hidden by $\Omega(1)$ in the parameter dependency is at most $\pi e < 8.6$ in all lower bounds.

**Optimal upper bound.** To complement our lower bound, we show that for strongly convex and strongly smooth functions, finding a solution to (1) can be done deterministically with

$$O\Big(Nd\kappa \log \kappa \log \frac{\beta d}{\varepsilon}\Big) \text{ total bits communicated,}$$

where $\sum_{i=1}^{N} f_i$ is $\alpha$-strongly convex and $\beta$-strongly smooth, and $\kappa = \beta/\alpha$ is the condition number. This algorithm matches our lower bound for constant condition number. It is a variant of deterministic quantised gradient descent (Magnússon et al., 2019). However, to achieve a tight bound, we need to (a) ensure that our gradient quantisation is sufficiently parsimonious, using $O(d \log \kappa)$ bits per gradient, and (b) avoid all-to-all exchange of gradients. For (a), we specialise a recent lattice-based quantisation scheme which allows arbitrary centering of iterates (Alistarh et al., 2020), and for (b), we use two-stage quantisation approach, where the nodes first send their quantised gradients to the coordinator, and the coordinator then broadcasts the carefully quantised sum back to nodes.

## 1.2 RELATED WORK

**Message-passing versus broadcast.** There are two communication models frequently used in both communication complexity and distributed optimisation: the first is the message-passing model we focus on, and the second is the *broadcast* or *blackboard* model, where each message sent by a node is seen by all nodes. The broadcast model is more powerful than the message-passing model: lower bounds for broadcast model apply also for message-passing, but upper bounds for broadcast do not directly translate to message passing. Arguably, message-passing model is closer to reality, as constant-cost, high-bandwidth broadcast mechanisms do not exist in real systems.

**Optimisation lower bounds.** The first communication lower bounds for a variant of (1) were given in the seminal work of Tsitsiklis & Luo (1987), who study optimising sums of convex functions in a two-machine setting. For *deterministic* algorithms, they prove that $\Omega(d \log(\beta_0 d/\varepsilon))$ bits are necessary. Zhang et al. (2013) use a nearly identical argument to give a $\Omega(d \log(\beta_0 d/\varepsilon))$ lower bound for randomised algorithms in the broadcast model. (See also Table 1 in the Appendix.)

The basic intuition behind these lower bounds is that a node without information about the input needs to receive $\Omega(d \log(\beta_0 d/\varepsilon))$ bits, as otherwise the node cannot produce sufficiently many different output distributions to cover all possible locations of the minimum (cf. Lemma 2.) It is worth emphasising that their bound is on the *received* bits of the *output node*, and does not directly imply anything for other nodes; for example, an algorithm where each node transmits $O\big((d \log(\beta_0 d/\varepsilon))/N\big)$ bits is not ruled out by these previous results. Generalising their approach to match our results seems challenging, as we would have to (a) explicitly require that all nodes output the solution, and (b) ensure that *no node* can use their local input as a source of extra information.

Our study was inspired in part by the recent work of Vempala et al. (2020), who characterised the communication complexity of solving linear systems, linear regression, and related problems, over bounded integer matrices. The results are based on communication complexity arguments, similarly to our lower bound. However, there are some notable technical differences: first, the arbitrary real functions we consider do not have a natural binary encoding, and therefore their approach would not directly extend to our setting. Second, the approximation ratio for linear regression is defined *multiplicatively* in their work, whereas we consider *additive* approximations. Both formulations are popular in terms of upper bounds, with the additive error formulation being arguably more popular in the context of machine learning applications. Overall, our results complement theirs, enriching the landscape of lower bounds for distributed optimisation.

**Statistical estimation lower bounds.** In *statistical estimation*, nodes receive random samples from some input distribution, and must infer properties of the input distribution, e.g. its mean. Specifically, for mean estimation, there are *statistical* limits on how good an estimate one can obtain from limited number of samples, although inputs are drawn from a distribution instead of adversarially. Concretely, the results of Shamir (2014) and Suresh et al. (2017) apply only to restricted types of protocols. Garg et al. (2014) and Braverman et al. (2016) give lower bounds for Gaussian mean estimation, where each node receives $s$ samples from a $d$-dimensional Gaussian distribution with variance $\sigma^2$. The latter reference shows that to achieve the minimax rate $\sigma^2 d/Ns$ on mean squared error requires $\Omega(Nd)$ total communication. These results do not imply optimal lower bounds for our setting.

**Lower bounds on round and oracle complexity.** Beyond bit complexity, one previous setting assumes that nodes can transmit vectors of real numbers, while restricting the types of computation allowed for the nodes. This is useful to establish bounds for the number of iterations required for convergence of distributed optimisation algorithms (Arjevani & Shamir, 2015; Scaman et al., 2017), but does not address the communication cost of a single iteration. A second related but different setting assumes the nodes can access their local functions only via specific oracle queries, such as *gradient or proximal queries*, and bound the number of such queries required to solve an optimisation problem (Woodworth & Srebro, 2016; Woodworth et al., 2018).

**Upper bounds.** There has been a tremendous amount of work recently on communication-efficient optimisation algorithms in the distributed setting. Due to space constraints, we focus on a small selection of closely-related work. One critical difference relative to practical references, e.g. Alistarh et al. (2017), is that they usually assume gradients are provided as 32-bit inputs, and focus on reducing the amount of communication *by constant factors*, which is reasonable in practice. One exception is Suresh et al. (2017), who present a series of quantisation methods for mean estimation on real-valued input vectors. Recently, Alistarh et al. (2020) studied the same problem, focusing on replacing the dependence on input norm with a *variance* dependence. We adapt their scheme for our upper bound.

Tsitsiklis & Luo (1987) gave a deterministic upper bound in a *two-node* setting, with $O\big(\kappa d \log(\kappa d) \log(\beta d/\varepsilon)\big)$ total communication cost. Recently, Magnússon et al. (2019) extended this to $N$-node case in the *broadcast* model, with $O\big(N\kappa d \log(\kappa d) \log(\beta d/\varepsilon)\big)$ total communication cost. For randomised algorithms and constant condition number, better upper bound of $O(Nd \log(\beta d/\varepsilon))$ total communication cost in the broadcast model follows by using QSGD stochastic quantisation (Alistarh et al., 2017) plugged into stochastic variance-reduced gradient descent (SVRG) (Johnson & Zhang, 2013). See Künstner (2017) for a detailed treatment. (See also Table 2 in the Appendix.)

## 2 PRELIMINARIES AND BACKGROUND

**Coordinator model.** We consider communication protocols in the classic *coordinator model* (Dolev & Feder, 1992; Phillips et al., 2012; Braverman et al., 2013). In this model, we have $N$ *nodes* as well as a separate *coordinator* node. The task is to compute the value of a function $\Gamma \colon B^N \to A$, where $B$ and $A$ are arbitrary input and output domains; each node $i = 1, 2, \ldots, N$ receives an input $b_i \in B$. There is a communication channel between each of the nodes and the coordinator, and nodes can communicate with the coordinator by exchanging binary messages. The coordinator has to output the value $\Gamma(b_1, b_2, \ldots, b_N)$. Furthermore, all nodes, including the coordinator, have access to a stream of private random bits.

More precisely, we assume without loss of generality that computation is performed as follows:

(1) Initially, each node $i = 1, 2, \ldots, N$ receives the input $b_i$. The coordinator and nodes $i = 1, 2, \ldots, N$ receive independent and uniformly random binary strings $r, r_i \in \{0, 1\}^c$, respectively, where $c$ is a constant.

(2) The computation then proceeds in sequential rounds, where in each round, (a) the coordinator first takes action by either outputting an answer, or sending a message to a single node $i$, and (b) the node $i$ that received a message from the coordinator responds by sending a a message to the coordinator.

A *transcript* for a node is a list of the messages it has sent and received. A *protocol* $\Pi$ is a mapping giving the actions of the coordinator and the nodes; for the coordinator, the next action is a function of its transcript so far and the private random bits $r$, and for node $i$, the next action is a function of its input $b_i$, its transcript so far and the private random bits $r_i$. The protocol $\Pi$ also determines the number of random bits the nodes receive.

We say that a protocol $\Pi$ computes $\Gamma \colon B^N \to A$ with error $p$ if for all $(b_1, b_2, \ldots, b_N) \in B^N$, the output of $\Pi$ is $\Gamma(b_1, b_2, \ldots, b_N)$ with probability at least $1 - p$. The *communication complexity* of a protocol $\Pi$ is the maximum number of total bits transmitted by all nodes, i.e. the total length of the transcripts, on any input $(b_1, b_2, \ldots, b_N) \in B^N$ and any private random bits of the nodes.

While the model definition may appear restrictive, the protocol restrictions do not matter when the complexity measure is the total number of bits exchanged. Any algorithm using parallel synchronous

or even asynchronous communication can be transformed into a sequential protocol by sequentialising the communication steps to occur one after the other. Likewise, algorithms using all-to-all message-passing can be transformed to the coordinator model, by routing all messages via the coordinator. The transformation incur at most constant factor overhead.

Finally, observe that the model is *nonuniform*, i.e. each protocol is defined only for specific functions $\Gamma \colon B^N \to A$ and specific input and output sets $B$ and $A$. As such, we do not need to impose any requirements on the computability of the protocol actions, rather these can be arbitrary functions. Any uniform algorithm working for a range of parameters induces a series of nonuniform protocols, so lower bounds for coordinator model translate to uniform algorithms.

**Communication complexity.**  We now recall some basic definitions and results from communication complexity. In the following, we assume that sets $B$ and $A$ are finite, as this is the standard setting of communication complexity.

For a function $\Gamma \colon B^N \to A$, the *deterministic communication complexity* $\mathsf{CC}(\Gamma)$ is the minimum communication complexity of a deterministic protocol computing $\Gamma$. Likewise, the *$\delta$-error randomised communication complexity* $\mathsf{RCC}^\delta(\Gamma)$ is the minimum communication complexity of a protocol that computes $\Gamma$ with error probability $\delta$.

For a distribution $\mu$ over $B^N$, we define the *$\delta$-error $\mu$-distributional communication complexity of $\Gamma$*, denoted by $\mathsf{D}_\delta^\mu(\Gamma)$, as the minimum communication complexity of a deterministic protocol that computes $\Gamma$ with error probability $\delta$ when the input is drawn from $\mu$. Similarly, the *$\delta$-error $\mu$-distributional expected communication complexity of $\Gamma$*, denoted by $\mathsf{ED}_\mu^\delta(\Gamma)$, is the minimum expected communication cost of a protocol that computes $\Gamma$ with error probability $\delta$, where the expectation is taken over input drawn from $\mu$ and the random bits of the protocol.

Yao's Lemma (Yao, 1977) relates the distributional communication complexity to the randomised communication complexity; see Woodruff & Zhang (2017) for a proof in the coordinator model.

**Lemma 1** (Yao's Lemma). *For function $\Gamma$ and $\delta > 0$, we have $\mathsf{RCC}^\delta(\Gamma) \geq \max_\mu \mathsf{D}_\mu^\delta(\Gamma)$.*

**Properties of convex functions.**  Recall that a continuously differentiable function $f$ is

$$\beta\text{-(strongly) smooth if} \qquad \|\nabla f(x) - \nabla f(y)\|_2 \leq \beta \|x - y\|_2 \,,$$
$$\alpha\text{-strongly convex if} \qquad \left(\nabla f(x) - \nabla f(y)\right)^T (x - y) \geq \alpha \|x - y\|_2^2$$

for all $x$ and $y$ in the domain of $f$. For $\alpha$-strongly convex and $\beta$-strongly smooth function $f$, we say that $f$ has *condition number* $\kappa = \beta/\alpha$. If $f_1$ is $\alpha_1$-strongly convex and $\beta_1$-strongly smooth and $f_2$ is $\alpha_2$-strongly convex and $\beta_2$-strongly smooth, then $f_1 + f_2$ is $(\alpha_1 + \alpha_2)$-strongly convex and $(\beta_1 + \beta_2)$-strongly smooth.

A quadratic function $f(x) = \beta \|x - y\|_2^2 + C$ is $\beta$-strongly convex and $\beta$-strongly smooth. For $\varepsilon > 0$, if $f(x) \leq \varepsilon$, then $\|x - x^*\|_2 \leq (\varepsilon/\beta)^{1/2}$. A sum of quadratics $F(x) = \sum_{j=1}^k a_j \|x - y_j\|_2^2$, where $y_j \in \mathbb{R}^d$ and $a_j \geq 0$ for $j = 1, 2, \ldots, k$, is a quadratic function $F(x) = A\|x - x^*\|_2^2 + C$, where $C$ is a constant and $x^* = \sum_{j=1}^k a_j y_j / A$ is the minimum of $F$.

**Point packing.**  We will make use of the following elementary result, which bounds the number of points we can pack into $[0, 1]^d$ while maintaining a minimum distance between all points.

**Lemma 2** (Tsitsiklis & Luo (1987)). *For $\delta > 0$ and $d \geq 1$, there is a set of points $S \subseteq [0, 1]^d$ with $\|x - y\|_2 > \delta$ for all distinct $x, y \in S$, and $|S| \geq (d^{1/2}/C\delta)^d$, where $C = (\pi e/2)^{1/2}$ is a constant.*

## 3  LOWER BOUNDS

### 3.1  DETERMINISTIC LOWER BOUND FOR QUADRATIC FUNCTIONS

We start with a warm-up result, proving a lower bound against deterministic protocols. Essentially, we show that even recognising if all the nodes have the same input function is hard. Recall that in the $N$-player equality over universe of size $d$, denoted by $\mathsf{EQ}_{d,N}$, each player $i$ is given an input $b_i \in \{0, 1\}^d$,

and the task is to decide if all players have the same input. That is, $\mathsf{EQ}_{d,N}(b_1, \ldots, b_N) = 1$ if all inputs are equal, and $0$ otherwise. It is known (Vempala et al., 2020) that the deterministic communication complexity of $\mathsf{EQ}_{d,N}$ is $\mathsf{CC}(\mathsf{EQ}_{d,N}) = \Omega(Nd)$.

**Theorem 3.** *Given parameters $N$, $d$, $\varepsilon$, $\beta_0$ and $\beta = \beta_0 N$ satisfying $d\beta/\varepsilon = \Omega(1)$, any deterministic protocol solving (1) for quadratic input functions $x \mapsto \beta_0 \|x - x_0\|_2^2$ has communication complexity*

$$\Omega\big(Nd\log(\beta d/\varepsilon)\big).$$

*Proof.* Assume $\Pi$ is a deterministic protocol solving (1) with communication complexity $C_\Pi$. We show that $\Pi$ can then solve $N$-party equality over a universe of size $D = \Omega(d \log(\beta d/\varepsilon))$, implying

$$C_\Pi = \Omega(ND) = \Omega\big(Nd\log(\beta d/\varepsilon)\big).$$

More specifically, let $S$ be the set given by Lemma 2 with $\delta = (\varepsilon/2\beta)^{1/2}$, and let $D = \lceil \log |S| \rceil = \Theta(d\log(\beta d/\varepsilon))$. Note that since we assume $d\beta/\varepsilon = \Omega(1)$, the set $S$ has at least two elements and $D \geq 1$. For technical convenience, assume $|S| = 2^D$, and identify each binary string $b \in \{0,1\}^D$ with an element $\tau(b) \in S$.

Next, assume that each node $i$ is given a binary string $b_i \in \{0,1\}^D$ as input, and we want to compute $\mathsf{EQ}_{D,N}(b_1, b_2, \ldots, b_N)$. The nodes simulate protocol $\Pi$ with input function $f_i$ for node $i$, where $f_i(x) = \beta_0 \|x - \tau(b_i)\|_2^2$. Let us denote $F = \sum_{i=1}^d f_i$. Upon termination of the protocol, the coordinator learns a point $y \in [0,1]^d$ satisfying $F(y) \leq F(x^*) + \varepsilon$ and an estimate $r \in \mathbb{R}$ satisfying $r \leq F(y) + \varepsilon$, where $x^*$ is the true global minimum. The coordinator can now adjudicate equality based on $F(y)$ as follows:

(1) If all inputs $b_i$ are equal, then the functions $f_i$ are also equal, and $F(x^*) = 0$. In this case, we have $F(y) \leq 2\varepsilon$, and the coordinator outputs 1.

(2) If there are nodes $i$ and $j$ such that $i \neq j$, then for all points $x \in [0,1]^d$, we have $f_i(x) + f_j(x) > 2\varepsilon$ by the definition of $S$, and thus $F(x^*) > 2\varepsilon$. In this case, we have $r > 2\varepsilon$, and the coordinator outputs 0.

Since communication is only used for the simulation of $\Pi$, this computes $\mathsf{EQ}_{D,N}(b_1, b_2, \ldots, b_N)$ with $C_\Pi$ total communication, completing the proof. $\square$

### 3.2 RANDOMISED LOWER BOUND FOR QUADRATIC FUNCTIONS

We now prove our main result, by giving a lower bound for communication complexity of any algorithm solving (1) that holds even for randomised protocols, albeit with a slightly weaker bound.

**Theorem 4.** *Given parameters $N$, $d$, $\varepsilon$, $\beta_0$ and $\beta = \beta_0 N$ satisfying $d\beta/N^2\varepsilon = \Omega(1)$, any protocol solving (1) for quadratic input functions $x \mapsto \beta_0 \|x - x_0\|_2^2$ has communication complexity*

$$\Omega\big(Nd\log(\beta d/N\varepsilon)\big).$$

**Discussion.** As this is our main result, we pause to discuss its implications. First, this result is more general than Theorem 3, as it applies to *stochastic* algorithms such as SGD or SVRG Johnson & Zhang (2013), which are arguably more popular in practice. The price for the increased generality is the additional $N$ factor in the denominator of the log, which appears due to technical requirements in the reduction. Finally, we note that the constant lower bound on $d\beta/N^2\varepsilon$ is the relatively small $e\pi < 8.6$, and that this lower bound is likely to hold in most practical settings of interest, as $d$ is usually quite large, and $\varepsilon$ is usually quite small.

**Proof overview.** To formally apply communication complexity tools, will prove a lower bound for a *discretised* version of (1) – where both the input and output sets are finite – which will imply Theorem 4. Let $N$, $d$, $\varepsilon$, and $\beta$ be fixed, assume $d\beta/N^2\varepsilon = \Omega(1)$, and

(1) let $S$ be the set given by Lemma 2 with $\delta = 3N(\varepsilon/\beta)^{1/2}$, and

(2) let $T \subseteq [0,1]^d$ be an arbitrary finite set of points such that for any $x \in [0,1]^d$, there is a point $t \in T$ with $\|x - t\| \leq (\varepsilon/4\beta)^{1/2}$.

By assumption $d\beta/N^2\varepsilon = \Omega(1)$, the set $S$ has size at least 2. Let $D = \lceil \log|S| \rceil = \Theta(d\log(\beta d/N\varepsilon))$. Again, for convenience, assume $2^D = |S|$, and identify each binary string $b \in \{0,1\}^D$ with an element $\tau(b) \in S$.

**Definition 5.** *Given parameters $N, d, \varepsilon, \beta$, we define the problem* $\mathsf{MEAN}_{d,N}^{\varepsilon,\beta}$ *as follows:*

- *The node inputs are from $\{0,1\}^D$, and*

- *Valid outputs for input $(b_1, b_2, \ldots, b_N)$ are points $t \in T$ that satisfy the condition $\|x^* - t\|_2 \leq (\varepsilon/\beta)^{1/2}$, where $x^* = \sum_{i=1}^N \tau(b_i)/N$ is the average over inputs.*

First, we observe that any algorithm for solving (1) can be used to solve $\mathsf{MEAN}_{d,N}^{\varepsilon,\beta}$.

**Lemma 6.** *For fixed $N, d, \varepsilon, \beta_0$ and $\beta = \beta_0 N$, any randomised protocol solving (1) for quadratic functions $x \mapsto \beta_0\|x - x_0\|_2^2$ with error probability $1/3$ has communication complexity at least* $\mathsf{RCC}^{1/3}(\mathsf{MEAN}_{d,N}^{\varepsilon,\beta/4})$.

The natural next step is to prove a lower bound on the communication complexity of $\mathsf{MEAN}_{d,N}^{\varepsilon,\beta}$. We do this by using an instance of the *symmetrisation technique* of Phillips et al. (2012), via reduction to the expected communication complexity of a *two-party* communication problem where one player has to learn the complete input of the other player. Specifically, in the two-player problem called $\mathsf{2\text{-}BITS}_d$, player 1 (Alice) receives a binary string $b \in \{0,1\}^d$, of length $d$, and the task is for player 2 (Bob) to output $b$. Let $\zeta_p$ be a distribution over binary strings $b \in \{0,1\}^d$ where each bit is set to 1 with probability $p$ and to 0 with probability $1-p$. The following lower bound for $\mathsf{2\text{-}BITS}_d$ is known, and holds even for protocols with *public randomness*, i.e. when Alice and Bob have access to the same string of random bits:

**Lemma 7** (Phillips et al. (2012))**.** $\mathsf{ED}_{\zeta_p}^{1/3}(\mathsf{2\text{-}BITS}_d) = \Omega(dp\log p^{-1})$.

**Lemma 8.** *For $N, d, \varepsilon$, and $\beta$ satisfying $d\beta/N^2\varepsilon = \Omega(1)$, we have*

$$\mathsf{RCC}^{1/3}(\mathsf{MEAN}_{d,N}^{\varepsilon,\beta}) = \Omega\left(N \cdot \mathsf{ED}_{\zeta_{1/2}}^{1/3}(\mathsf{2\text{-}BITS}_D)\right) = \Omega(Nd\log(\beta d/N\varepsilon)).$$

Due to space constraints, we defer the proofs of Lemmas 6 and 8 to Appendix A.1. Theorem 4 now follows immediately from Lemmas 6 and 8. The result can be generalised for arbitrary convex domains $\mathbb{D} \subseteq \mathbb{R}^d$ as $\Omega(N\log s)$, given a point packing bound $s$ for $\mathbb{D}$ as in Lemma 2.

### 3.3 LOWER BOUND FOR NON-CONVEX FUNCTIONS

We now show a simple lower bound for optimisation over non-convex objective functions. Specifically, we construct a set of hard input functions as follows. Let $\varepsilon$, $d$ and $\beta$ be constant satisfying $d\beta/\varepsilon = \Omega(1)$, and consider the set $S$ given by Lemma 2 with $\delta = 2\varepsilon/\beta$. This gives a set $S$ with size at least $(\beta d^{1/2}/2C\varepsilon)^d = \exp(\Omega(d\log(\beta d)/\varepsilon))$. Let us identify the points in $S$ with elements of $\{1, 2, \ldots, |S|\}$. For a binary string $b \in \{0,1\}^{|S|}$, define the function $f_b$ by

$$f_b(x) = \begin{cases} \beta\|x - s\|_2 & \text{if } \|x - s\|_2 < \varepsilon/\beta \text{ for } s \text{ with } b_s = 1, \\ \varepsilon & \text{otherwise.} \end{cases}$$

Since the distance between points in $S$ is at least $2\varepsilon/\beta$, the functions $f_T$ are well-defined, continuous and $\beta$-Lipschitz. The proof works by reduction from $N$-player set disjointness (Braverman et al., 2013); we defer the details to Appendix B.

**Theorem 9.** *Given parameters $N$, $d$, $\varepsilon$ and $\beta$ satisfying $d\beta/\varepsilon = \Omega(1)$ and $(\beta d^{1/2}/2C\varepsilon)^d = \omega(\log N)$, any protocol solving (1) with error probability $\delta > 0$ when the inputs are guaranteed to be functions $f_b$ for $b \in \{0,1\}^{|S|}$ has communication complexity $N\exp(\Omega(d\log(\beta d)/\varepsilon))$.*

## 4 TIGHT DETERMINISTIC UPPER BOUND

We now present a *deterministic* algorithm which matches our lower bound for constant condition number, in the coordinator model. We follow the general structure of communication-reduced algorithms, e.g. Magnússon et al. (2019): the nodes collectively execute an instance of gradient descent

(GD), where they carefully *quantise* their updates, accumulated at a coordinator. The algorithm relies on two new technical ingredients to achieve optimality: 1) we use two-step quantisation to avoid (inherently suboptimal) all-to-all communication; 2) we remove a superfluous $\log d$ factor in the communication by employing a lattice-based quantisation scheme allowing for arbitrary centring of the gradient estimates to be averaged (Alistarh et al., 2020). The second step causes non-trivial complications, as this scheme may *fail* if inputs are too far apart.

**Preliminaries.** We assume that the input functions of each node $i$ is $f_i \colon \{0,1\}^d \to \mathbb{R}$, which is $\alpha_0$-strongly convex and $\beta_0$-strongly smooth. This implies that $F = \sum_{i=1}^{N}$ is $\alpha$-strongly convex and $\beta$-strongly smooth for $\alpha = N\alpha_0$ and $\beta = N\beta_0$. Consequently, the functions $f_i$ and $F$ have condition number bounded by $\kappa = \beta/\alpha$. Furthermore, we assume that the local functions $f_i$ all have minimum value $\inf_{x \in [0,1]^d} f_i(x) = 0$, and thus range $[0, \beta_0 d]$.

We aim to reach the global minimum $x^\star$ of the sum $\sum_{i=1}^{N} f_i(x)$ by starting from an arbitrary point $x^0 \in [0,1]^d$, and applying a surrogate of the GD update

$$ x^{(t+1)} = x^{(t)} - \gamma \sum_{i=1}^{N} \nabla f_i(x^{(t)}), \text{ where } \gamma > 0 \text{ is the learning rate parameter.} $$

It is well-known, e.g. Bubeck (2015), that GD converges at an exponential rate in $(1 - 1/\kappa)$.

The algorithm has each node generate gradients of its local function $f_i$, and quantise them in a carefully-parametrised way. Specifically, the quantisation we use works in a setting where the nodes all know a point $q \in [0,1]^d$, and the points to be quantised are in the vicinity of $q$ – in the algorithm, the point $q$ will be the previous quantised gradient. The quantisation is parametrised by $R$, the maximum distance between the input point and $q$, and by $\varepsilon$, the maximum quantisation error we wish to tolerate.

**Corollary 10** (Alistarh et al. (2020)). *Let $R$ and $\varepsilon$ be fixed positive parameters, and $q \in \mathbb{R}^d$ be an estimate vector, and $B \in \mathbb{N}$ be the number of bits used by the quantisation scheme. Then, there exists a deterministic quantisation scheme, specified by a function $Q_{\varepsilon,R} \colon \mathbb{R}^d \times \mathbb{R}^d \to \mathbb{R}^d$, an encoding function $\mathrm{enc}_{\varepsilon,R} \colon \mathbb{R}^d \to \{0,1\}^B$, and a decoding function $\mathrm{dec}_{\varepsilon,R} \colon \mathbb{R}^d \times \{0,1\}^B \to \mathbb{R}^d$, with the following properties:*

*(1) (Validity.) $\mathrm{dec}_{\varepsilon,R}(q, \mathrm{enc}_{\varepsilon,R}(x)) = Q_{\varepsilon,R}(x,q)$ for all $x, q \in \mathbb{R}^d$ with $\|x - q\|_2 \le R$.*

*(2) (Accuracy.) $\|Q_{\varepsilon,R}(x,q) - x\|_2 \le \varepsilon$ for all $x, q \in \mathbb{R}^d$ with $\|x - q\|_2 \le R$.*

*(3) (Cost.) If $\varepsilon = \lambda R$ for any $\lambda < 1$, the bit cost of the scheme satisfies $B = O(d \log \lambda^{-1})$.*

**Algorithm description.** We now describe the algorithm, and overview its guarantees. The full description and analysis are available in Appendix C. We assume that the constants $\alpha$ and $\beta$ are known to all nodes, so the parameters of the quantised gradient descent can be computed locally, and use $W$ to be an upper bound on the diameter on the convex domain $\mathbb{D}$, e.g. $W = d^{1/2}$ if $\mathbb{D} = [0,1]^d$. We assume that the initial iterate $x^{(0)}$ is arbitrary, but the same at all nodes, and set the initial quantisation estimates $q^{(0)}$ and $q_i^{(0)}$ at each $i$ as the origin.

The algorithm proceeds in rounds $t = 1, 2, \ldots, T$. At the beginning of round $t + 1$, each node $i$ knows the values of the iterate $x^{(t)}$, the previous global quantised gradient $q^{(t)}$, and its local quantised gradient $q_i^{(t)}$; the coordination knows all these values. We define the following parameters for the algorithm. Let $\gamma = \beta^{-1}$ and $\xi = (1 - \kappa^{-1})$ be the step size and convergence rate of gradient descent, and let $W$ be such that $\|x^0 - x^*\| \le W$. We define

$$ K = 2/\xi, \qquad \delta = \xi(1 - \xi)/4, \qquad \mu = \delta K + \xi, \qquad R^{(t)} = \beta K W \mu^t. $$

Assuming $\kappa \ge 2$, we have $\mu < 1$, $\xi \ge 1/2$ and $K \ge 1$. At step $t$, nodes perform the following steps:

(1) Each node $i$ updates its iterate as $x^{(t+1)} = x^{(t)} - \gamma q^{(t)}$.

(2) Each node $i$ computes its local gradient over $x^{(t+1)}$, and transmits it in quantised form to the coordinator as follows. Let $\varepsilon_1 = \delta R^{(t+1)}/(2N)$ and $\rho_1 = R^{(t+1)}/N$.

(a) Node $i$ computes $\nabla f_i(x^{(t+1)})$ locally, and sends message $m_i = \text{enc}_{\varepsilon_1, \rho_1}(\nabla f_i(x^{(t+1)}))$ to the coordinator.

(b) The coordinator receives messages $m_i$ for $i = 1, 2, \ldots, N$, and decodes them as $q_i^{(t+1)} = \text{dec}_{\varepsilon_1, \rho_1}(q_i^{(t)}, m_i)$. The coordinator then computes $r^{(t+1)} = \sum_{i=1}^{N} q_i^{(t+1)}$.

(3) The coordinator sends the quantised sum of gradients to all other nodes as follows. Let $\varepsilon_2 = \delta R^{(t+1)}/2$ and $\rho_2 = (1 + \delta/2)R^{(t+1)}$.

(a) The coordinator sends the message $m = \text{enc}_{\varepsilon_2, \rho_2}(r^{(t+1)})$ to each node $i$.

(b) Each node decodes the coordinator's message as $q^{(t+1)} = \text{dec}_{\varepsilon_2, \rho_2}(q^{(t)}, m)$.

**Guarantees.** The key technical trick behind the algorithm is the extremely careful choice of parameters for quantisation at every step. This balances the fact that the quantisation has to be fine enough to ensure optimal GD convergence, but coarse enough to ensure optimal communication cost. Overall, the algorithm ensures the following guarantees, whose proof is provided in Appendix C.

**Theorem 11.** *Let $\varepsilon > 0$, a dimension $d$, and a convex domain $\mathbb{D} \subseteq \mathbb{R}^d$ of diameter $W$ be fixed. Given $N$ nodes, each assigned a function $f_i \colon \mathbb{D} \to \mathbb{R}$ such that $F = \sum_{i=1}^{N} f_i$ is $\alpha$-strongly convex and $\beta$-smooth, the above algorithm converges to a point $x^{(T)}$ with $\|F(x^{(T)}) - F(x^*)\|_2 \leq \varepsilon$ using*

$$O\left(Nd\kappa \log \kappa \log \frac{\beta W}{\varepsilon}\right) \text{ bits of communication.}$$

## 5 DISCUSSION AND FUTURE WORK

We have provided the first tight bounds on the communication complexity of optimising sums of quadratic functions in the $N$-party model with a coordinator. Our results are algorithm-independent, and immediately imply the same lower bound for the practical parameter server and decentralised models of distributed optimisation.

In terms of future work, we expect that the randomised lower bound could be improved to match the deterministic one even for small $d$, possibly via reduction from a suitable *gap problem* in communication complexity (e.g. Chakrabarti & Regev (2011)). Another avenue for future work is to investigate tight upper and lower bounds in the case where the functions being optimised are not quadratics, as isolating the "right" dependency on the condition number does not appear immediate. Finally, understanding the exact complexity of optimisation in the *broadcast model* remains open.

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

Table 1: Comparison of existing lower bounds on total communication required to solve (1). 'BC' denotes results for broadcast model, and 'MP' denotes results for message-passing model. Note that lower bounds for the broadcast model also apply to the message-passing model.

| Problem | Lower bound | Model | | |
|---|---|---|---|---|
| Quadratic optimisation | $\Omega(d \log \frac{\beta d}{\varepsilon})$ | 2-node | Det. | Tsitsiklis & Luo (1987) |
| | $\Omega(d \log \frac{\beta d}{\varepsilon})$ | BC | Rand. | Garg et al. (2014) |
| | $\Omega(N d \log \frac{\beta d}{\varepsilon})$ | MP | Det. | §3.1 |
| | $\Omega(N d \log \frac{\beta d}{N\varepsilon})$ | MP | Rand. | §3.2 |
| Gaussian mean estimation | $\Omega(Nd/\log N)$ | BC | Rand. | Garg et al. (2014) |
| | $\Omega(Nd)$ | BC | Rand. | Braverman et al. (2016) |

Table 2: Upper bounds for distributed optimisation over $\beta$-smooth, $\alpha$-strongly convex input functions with condition number $\kappa$. 'BC' denotes results for broadcast model, and 'MP' denotes results for message-passing model. Note that upper bounds for the message-passing model also apply to the broadcast model, but not vice versa.

| Input functions | Upper bound | Model | | |
|---|---|---|---|---|
| Constant $\kappa$ | $O(N d \log \frac{\beta d}{\varepsilon})$ | BC | Rand. | Alistarh et al. (2017); Künstner (2017) |
| General | $O(\kappa d \log(\kappa d) \log \frac{\beta d}{\varepsilon})$ | 2-node | Det. | Tsitsiklis & Luo (1987) |
| | $O(N \kappa d \log(\kappa d) \log \frac{\beta d}{\varepsilon})$ | BC | Det | Magnússon et al. (2019) |
| | $O(N d \kappa \log \kappa \log \frac{\beta d}{\varepsilon})$ | MP | Det. | §4 |

# A   OMITTED PROOFS, SECTION 3.2

## A.1   PROOF OF LEMMA 6

*Proof of Lemma 6.* Let $\Pi$ be protocol solving (1) with communication complexity $C$ and error probability $1/3$. We show that we can use it to solve $\mathsf{MEAN}_{d,N}^{\varepsilon,\beta/4}$ with total communication cost $C$ and error probability $1/3$, implying the claim. Given input $(b_1, b_2, \ldots, b_N)$ for $\mathsf{MEAN}_{d,N}^{\varepsilon,\beta/4}$, nodes can simulate the protocol $\Pi$ with input functions $f_i(x) = \beta_0 \|x - \tau(b_i)\|_2^2$. By the properties of quadratic functions, we have $F(x) = \sum_{i=1}^{N} f_i(x) = \beta \|x - x^*\|_2^2 + C$, where $x^* = \sum_{i=1}^{N} \frac{\tau(b_i)}{N}$. Thus, the output $y$ of $\Pi$ satisfies $\|y - x^*\|_2 \le (\varepsilon/\beta)^{1/2}$. The coordinator now outputs the closest point $t \in T$ to $y$. We therefore have

$$\|x^* - t\|_2 = \|x^* - y + y - t\|_2 \le \|x^* - y\|_2 + \|y - t\|_2 \le 2(\varepsilon/\beta)^{1/2} = (4\varepsilon/\beta)^{1/2}.$$

$\square$

## A.2   PROOF OF LEMMA 8

*Proof of Lemma 8.* Let $\mu$ denote a distribution on $\prod_{i=1}^{N} \{0,1\}^D$, where each $D$-bit string is selected uniformly at random, and let $\zeta$ be uniformly random on $\{0,1\}^D$. We will prove that

$$\mathsf{D}_\mu^{1/3}(\mathsf{MEAN}_{d,N}^{\varepsilon,\beta}) = \Omega\big(N \cdot \mathsf{ED}_\zeta^{1/3}(\mathsf{2\text{-}BITS}_D)\big).$$

Since $\mathsf{ED}_\zeta^{1/3}(\mathsf{2\text{-}BITS}_D) = \Omega(D)$ by Lemma 7, the claim follows by Yao's Lemma.

Suppose now that we have a deterministic protocol $\Pi_1$ for $\mathsf{MEAN}_{d,N}^{\varepsilon,\beta}$ with worst-case communication cost $C$ and error probability $1/3$ on input distribution $\mu$. Given $\Pi_1$, we define a 2-player protocol $\Pi_2$ with public randomness for $\mathsf{2\text{-}BITS}_D$ as follows; assume that Alice is given $b \in \{0,1\}^D$ as input.

(1) Alice and Bob pick a random index $i \in [N]$ uniformly at to select a random $i$ node using the shared randomness. Without loss of generality, we can assume that the picked node was node $i = 1$.

(2) Alice and Bob simulate protocol $\Pi_1$, with Alice simulating node 1 and Bob simulating the coordinator and nodes $2, 3, \ldots, N$. For the inputs $b_1, b_2, \ldots, b_N$ to $\Pi_1$, Alice sets $b_1 = x$, and Bob selects the inputs $b_2, b_3, \ldots, b_N$ uniformly at random by using the public randomness. Messages $\Pi_1$ sends between the coordinator and node 1 are communicated between Alice and Bob, and all other communication is simulated by Bob internally.

(3) Once the simulation is complete, Bob knows the output $t \in T$ of $\Pi_1$ which satisfies $\|t - z^*\|_2 \leq (\varepsilon/\beta)^{1/2}$, where $z^* = \sum_{i=1}^N \tau(b_i)/N$.

As the final step, we show that Bob can now recover Alice's input from $t$. Let $y = \sum_{i=2}^N \tau(b_i)/(N-1)$ be the weighted average of points $\tau(b_2), \tau(b_3), \ldots, \tau(b_N)$. We now have that $Nz^* - (N-1)y = \tau(b_1)$ by simple calculation.

Since $\|t - z^*\|_2 \leq (\varepsilon/\beta)^{1/2}$, it follows that

$$\begin{aligned}
\|(Nt - (N-1)y) - \tau(b_1)\|_2 &= \|Nt - (N-1)y - Nz^* + Nz^* - \tau(b_1)\|_2 \\
&= \|Nt - Nz^* + Nz^* - (N-1)y - \tau(b_1)\|_2 \\
&= \|Nt - Nz^*\|_2 = N\|t - z^*\|_2 \leq N(\varepsilon/\beta)^{1/2}\,.
\end{aligned}$$

Since the distance between any two points in $S$ is at least $3N(\varepsilon/\beta)^{1/2}$, we have that $\tau(b_1)$ is the only point from $S$ within distance $(\varepsilon/\beta)^{1/2}$ from $Nz - (N-1)y$. As Bob knows both $z$ and $\tau(b_2), \tau(b_3), \ldots, \tau(b_N)$ after the simulation, he can recover the point $x_1$ and thus infer Alice's input.

Now let us analyse the expected cost of $\Pi_2$ under input distribution $\zeta$. First, observe that since the simulation runs $\Pi_1$ on input distribution $\mu$, the output $y$ is correct with probability $2/3$, and thus the output of $\Pi_2$ is correct with probability $2/3$. Now let $C_{\Pi_1}$ be the worst-case communication cost of $\Pi_1$ and let $C_{\Pi_1}(b_1, \ldots, b_N)$ and $C_{\Pi_1,i}(b_1, \ldots, b_N)$ denote the total communication cost and the communication used by node $i$ in $\Pi_1$ on input $b_1, \ldots, b_N$, respectively. Finally, let $C_{\Pi_2}(b,r)$ be a random variable giving the communication cost of $\Pi_2$ on input $b$ and random bits $r$.

Now we have that

$$\begin{aligned}
\mathbb{E}_{b_1,r}[C_{\Pi_2}(b_1,r)] &= \sum_{b_1 \in \{0,1\}^D} \frac{1}{2^D} \mathbb{E}_r[C_{\Pi_2}(b_1,r)] \\
&= \sum_{b_1 \in \{0,1\}^D} \frac{1}{2^D} \sum_{b_2,\ldots,b_N} \sum_{i=1}^N \frac{C_{\Pi_1,i}(b_1,\ldots,b_N)}{N2^{(N-1)D}} \\
&= \frac{1}{N} \sum_{b_1,b_2,\ldots,b_N} \frac{1}{2^{ND}} \sum_{i=1}^N C_{\Pi_1,i}(b_1,\ldots,b_N) \\
&= \frac{1}{N} \sum_{b_1,b_2,\ldots,b_N} \frac{1}{2^{ND}} C_{\Pi_1}(b_1,b_2,\ldots,b_N) \\
&\leq \frac{1}{N} \sum_{b_1,b_2,\ldots,b_N} \frac{1}{2^{ND}} C_{\Pi_1} = \frac{C_{\Pi_1}}{N}
\end{aligned}$$

Since $\mathbb{E}_{b_1,\beta}[C_{\Pi_2}(b_1,r)] \geq \mathsf{ED}_\zeta^{1/3}(\mathsf{2\text{-}BITS}_D)$, and the argument holds for any protocol $\Pi_1$ solving $\mathsf{MEAN}_{d,N}^{\varepsilon,\beta}$ with error probability $1/3$, we have that

$$\mathsf{D}_\mu^{1/3}(\mathsf{MEAN}_{d,N}^{\varepsilon,\beta}) \geq N \cdot \mathsf{ED}_\zeta^{1/3}(\mathsf{2\text{-}BITS}_D)\,,$$

completing the proof. $\qquad\square$

## B    LOWER BOUND FOR NON-CONVEX FUNCTIONS, FULL VERSION

We now show a simple lower bound for optimisation over non-convex objective functions. We reduce from the $N$-player *set disjointness* over universe of size $d$, denoted by $\mathsf{DISJ}_{d,N}$: each player $i$ is given an input $b_i \in \{0,1\}^d$, and the coordinator needs to output 0 if there is a coordinate $\ell \in [d]$ such that $b_i(\ell) = 1$ for all $i \in [N]$, and 1 otherwise.

**Theorem 12** (Braverman et al. (2013))**.** *For $\delta > 0$, $N \geq 1$ and $d = \omega(\log N)$, the randomised communication complexity of set disjointness is* $\mathsf{RCC}^\delta(\mathsf{DISJ}_{d,N}) = \Omega(Nd)$.

Again consider for fixed $\varepsilon$, $d$ and $\beta$ the set $S$ given by Lemma 2 with $\delta = 2\varepsilon/\beta$. This gives a set $S$ with size at least $(\beta d^{1/2}/2C\varepsilon)^d = \exp(\Omega(d\log(\beta d)/\varepsilon)$. Let us identify the points in $S$ with indices in $[|S|]$. For a binary string $b \in \{0,1\}^{|S|}$, define the function $f_b$ by

$$f_b(x) = \begin{cases} \beta\|x - s\|_2 & \text{if } \|x - s\|_2 < \varepsilon/\beta \text{ for } s \text{ with } b_s = 1, \\ \varepsilon & \text{otherwise.} \end{cases}$$

Since the distance between points in $S$ is at least $2\varepsilon/\beta$, the functions $f_b$ are well-defined, continuous and $\beta$-Lipschitz.

**Theorem 9.** *Given parameters $N$, $d$, $\varepsilon$ and $\beta$ satisfying $d\beta/\varepsilon = \Omega(1)$ and $(\beta d^{1/2}/2C\varepsilon)^d = \omega(\log N)$, any protocol solving (1) with error probability $\delta > 0$ when the inputs are guaranteed to be functions $f_b$ for $b \in \{0,1\}^{|S|}$ has communication complexity $N \exp(\Omega(d\log(\beta d)/\varepsilon))$.*

*Proof.* Assume there is a protocol $\Pi$ with the properties stated in the claim, and worst-case communication cost $C_\Pi$. We now show that we can use $\Pi$ to solve set disjointness over universe of size $|S|$ with $C_\Pi$ total communication, which implies

$$C_\Pi \geq \mathsf{RCC}^\delta(\mathsf{DISJ}_{|S|,N}) = \Omega(N \exp(\Omega(d\log(\beta d)/\varepsilon))),$$

yielding the claim.

First, we observe that for $b_1, b_2, \ldots b_N \in \{0,1\}^{|S|}$ that all contain 1 in some position $s$, then we have $\sum_{i=1}^N f_{b_i}(x) = 0$. Otherwise, for any point $x \in [0,1]^d$, consider the closest point $s \in S$ to $x$; there is at least one $b_i$ with $b_s = 0$, and for that function $f_{b_i}(x) = \varepsilon$ by definition. Thus, if $b_1, b_2, \ldots b_N$ are a YES-instance for set disjointness, then $\inf_{x \in [0,1]^d} \sum_{i=1}^N f_{b_i}(x) \geq \varepsilon$, and if $b_1, b_2, \ldots b_N$ are a NO-instance, then $\inf_{x \in [0,1]^d} \sum_{i=1}^N f_{b_i}(x) = 0$.

By definition, $\Pi$ can be used to distinguish between the two cases, and thus to solve set disjointness. $\square$

## C    DESCRIPTION AND ANALYSIS OF THE UPPER BOUND, FULL VERSION

We now describe in detail our deterministic upper bound. Our algorithm uses quantised gradient descent, loosely following the outline of Magnússon et al. (2019). However, there are two crucial differences. First, we use a carefully-calibrated instance of the quantisation scheme of Alistarh et al. (2020) to remove a $\log d$ factor from the communication cost, and second, we use use two-step quantisation to avoid all-to-all communication.

**Preliminaries on gradient descent.**    We will assume that the input functions $f_i \colon [0,1]^d \to \mathbb{R}$ are $\alpha_0$-strongly convex and $\beta_0$-strongly smooth. This implies that $F = \sum_{i=1}^N f_i$ is $\alpha$-strongly convex and $\beta$-strongly smooth for $\alpha = N\alpha_0$ and $\beta = N\beta_0$. Consequently, the functions $f_i$ and $F$ have condition number bounded by $\kappa = \beta/\alpha$. Furthermore, we assume that the local functions $f_i$ all have minimum value $\inf_{x \in [0,1]^d} f_i(x) = 0$, and thus range $[0, \beta_0 d]$.

*Gradient descent* optimises the sum $\sum_{i=1}^N f_i(x)$ by starting from an arbitrary point $x^{(0)} \in [0,1]^d$, and applying the update rule

$$x^{(t+1)} = x^{(t)} - \gamma \sum_{i=1}^N \nabla f_i(x^{(t)}),$$

where $\gamma > 0$ is a parameter.

Let $x^*$ denote the global minimum of $F$. We use the following standard result on the convergence of gradient descent; see e.g. Bubeck (2015).

**Theorem 13.** *For $\gamma = \beta^{-1}$, we have that $\|x^{(t+1)} - x^*\|_2 \leq (1 - \kappa^{-1})\|x^{(t)} - x^*\|_2$.*

**Preliminaries on quantisation.** For compressing the gradients the nodes will send to coordinator, we use the recent quantisation scheme of Alistarh et al. (2020). Whereas the original uses randomised selection of the quantisation point to obtain a unbiased estimator, we can use a deterministic version that picks an arbitrary feasible quantisation point (e.g. the closest one). This gives the following guarantees:

**Corollary 10** (Alistarh et al. (2020)). *Let $R$ and $\varepsilon$ be fixed positive parameters, and $q \in \mathbb{R}^d$ be an estimate vector, and $B \in \mathbb{N}$ be the number of bits used by the quantisation scheme. Then, there exists a deterministic quantisation scheme, specified by a function $Q_{\varepsilon,R}\colon \mathbb{R}^d \times \mathbb{R}^d \to \mathbb{R}^d$, an encoding function $\mathrm{enc}_{\varepsilon,R}\colon \mathbb{R}^d \to \{0,1\}^B$, and a decoding function $\mathrm{dec}_{\varepsilon,R}\colon \mathbb{R}^d \times \{0,1\}^B \to \mathbb{R}^d$, with the following properties:*

*(1) (Validity.) $\mathrm{dec}_{\varepsilon,R}(q, \mathrm{enc}_{\varepsilon,R}(x)) = Q_{\varepsilon,R}(x,q)$ for all $x, q \in \mathbb{R}^d$ with $\|x - q\|_2 \leq R$.*

*(2) (Accuracy.) $\|Q_{\varepsilon,R}(x,q) - x\|_2 \leq \varepsilon$ for all $x, q \in \mathbb{R}^d$ with $\|x - q\|_2 \leq R$.*

*(3) (Cost.) If $\varepsilon = \lambda R$ for any $\lambda < 1$, the bit cost of the scheme satisfies $B = O(d \log \lambda^{-1})$.*

### C.1 ALGORITHM DESCRIPTION

We now describe the algorithm, and overview its guarantees. We assume that the constants $\alpha$ and $\beta$ are known to all nodes, so the parameters of the quantised gradient descent can be computed locally, and use $W$ to be an upper bound on the diameter on the convex domain $\mathbb{D}$, e.g. $W = d^{1/2}$ if $\mathbb{D} = [0,1]^d$. We assume that the initial iterate $x^{(0)}$ is arbitrary, but the same at all nodes, and set the initial quantisation estimate $q_i^{(0)}$ at each $i$ as the origin.

The algorithm proceeds in rounds $t = 1, 2, \ldots, T$. At the beginning of round $t + 1$, each node $i$ knows the values of the iterate $x^{(t)}$, the global quantisation estimate $q^{(t)}$, and its local quantisation estimate $q_i^{(t)}$ for $i = 1, 2, \ldots, N$. We define the following parameters for the algorithm. Let $\gamma = \beta^{-1}$ and $\xi = (1 - \kappa^{-1})$ be the step size and convergence rate of gradient descent, and let $W$ be such that $\|x^{(0)} - x^*\| \leq W$. We define

$$K = 2/\xi, \qquad \delta = \xi(1-\xi)/4, \qquad \mu = \delta K + \xi, \qquad R^{(t)} = \beta K W \mu^t .$$

Assuming $\kappa \geq 2$, we have $\mu < 1$, $\xi \geq 1/2$ and $K \geq 1$. At step $t$, nodes perform the following steps:

(1) Each node $i$ updates its iterate as $x^{(t+1)} = x^{(t)} - \gamma q^{(t)}$.

(2) Each node $i$ computes its local gradient over $x^{(t+1)}$, and transmits it in quantised form to the coordinator as follows. Let $\varepsilon_1 = \delta R^{(t+1)}/(2N)$ and $\rho_1 = R^{(t+1)}/N$.

   (a) Node $i$ computes $\nabla f_i(x^{(t+1)})$ locally, and sends message $m_i = \mathrm{enc}_{\varepsilon_1,\rho_1}(\nabla f_i(x^{(t+1)}))$ to the coordinator.

   (b) The coordinator receives messages $m_i$ for $i = 1, 2, \ldots, N$, and decodes them as $q_i^{(t+1)} = \mathrm{dec}_{\varepsilon_1,\rho_1}(q_i^{(t)}, m_i)$. The coordinator then computes $r^{(t+1)} = \sum_{i=1}^{N} q_i^{(t+1)}$.

(3) The coordinator sends the quantised sum of gradients to all other nodes as follows. Let $\varepsilon_2 = \delta R^{(t+1)}/2$ and $\rho_2 = (1 + \delta/2)R^{(t+1)}$.

   (a) The coordinator sends the message $m = \mathrm{enc}_{\varepsilon_2,\rho_2}(r^{(t+1)})$ to each node $i$.

   (b) Each node decodes the coordinator's message as $q^{(t+1)} = \mathrm{dec}_{\varepsilon_2,\rho_2}(q^{(t)}, m)$.

After round $T$, all nodes know the final iterate $x^{(T)}$. The nodes compute their local value $f_i(x^{(T)})$, and send an approximate value to the coordinator; specifically, each node computes a partition of the range $[0, \beta_0 d]$ into segments of length $\varepsilon/N$, and sends the index of the smallest segment endpoint $r$ satisfying $r \geq f_i(x^{(T)})$ to the coordinator.

### C.2 ANALYSIS

For simplicity, we will split the analysis into two parts. The first describes and analyses the algorithm in an abstract way; the second part describes the details of implementing it in the coordinator model. For technical convenience, assume $\kappa \geq 2$; for smaller condition numbers, we can run the algorithm with $\kappa = 2$.

**Convergence.** Let $\gamma = \beta^{-1}$, let $x^{(0)} \in [0,1]^d$, $q_i^{(0)} \in \mathbb{R}^d$ and $q_i^{(0)} \in \mathbb{R}^d$ for $i = 1, 2, \ldots, N$ be arbitrary initial values. From the algorithm description, we see that the update rule for our quantised gradient descent is

$$x^{(t+1)} = x^{(t)} - \gamma q^{(t)},$$

$$q_i^{(t+1)} = Q\big(\nabla f_i(x^{(t+1)}), q_i^{(t)}, R^{(t+1)}/N, \delta R^{(t+1)}/(2N)\big),$$

$$r^{(t+1)} = \sum_{i=1}^{N} q_i^{(t+1)},$$

$$q^{(t+1)} = Q\big(r^{(t+1)}, q^{(t)}, (1 + \delta/2)R^{(t+1)}, \delta R^{(t+1)}/2\big).$$

**Lemma 14.** *The inequalities*

$$\|x^{(t)} - x^*\|_2 \leq \mu^t W, \tag{Q1}$$

$$\|\nabla f_i(x^{(t)}) - q_i^{(t)}\|_2 \leq \delta R^{(t)}/(2N), \tag{Q2}$$

$$\|\nabla F(x^{(t)}) - q^{(t)}\|_2 \leq \delta R^{(t)} \tag{Q3}$$

*hold for all $t$, assuming that they hold for $x^{(0)}$, $q^{(0)}$ and $q_i^{(0)}$ for $i = 1, 2, \ldots, N$.*

*Proof.* We apply induction over $t$; we assume that all the inequalities hold for $t$, and prove that they also hold for $t + 1$. Since we assume the inequalities hold for $t = 0$, the base case is trivial.

*Convergence (Q1)*: We have

$$\begin{aligned}
\|x^{(t+1)} - x^*\|_2 &= \|x^{(t)} - \gamma q^{(t)} + \gamma \nabla F(x^{(t)}) - \gamma \nabla F(x^{(t)}) + x^*\|_2 \\
&\leq \|\gamma q^{(t)} - \gamma \nabla F(x^{(t)})\|_2 + \|(x^{(t)} - \gamma \nabla F(x^{(t)})) - x^*\|_2 \\
&\leq \gamma \|\nabla F(x^{(t)}) - q^{(t)}\|_2 + \xi \|x^{(t)} - x^*\|_2 \\
&\leq \beta^{-1} \delta R^{(t)} + \xi \mu^t W \\
&= \beta^{-1} \delta \beta K \mu^t W + \xi \mu^t W \\
&= (\delta K + \xi) \mu^t W = \mu^{t+1} W.
\end{aligned}$$

*Local quantisation (Q2)*: First, let us observe that to prove that (Q2) holds for $t + 1$, it is sufficient to show $\|\nabla f_i(x^{(t+1)}) - q_i^{(t)}\|_2 \leq R^{(t+1)}/N$, as the claim then follows from the definition of $q_i^{(t+1)}$ and Corollary 10. We have

$$\begin{aligned}
\|\nabla f_i(x^{(t+1)}) - q_i^{(t)}\|_2 &= \|\nabla f_i(x^{(t+1)}) - \nabla f_i(x^{(t)}) + \nabla f_i(x^{(t)}) - q_i^{(t)}\|_2 \\
&\leq \|\nabla f_i(x^{(t+1)}) - \nabla f_i(x^{(t)})\|_2 + \|\nabla f_i(x^{(t)}) - q_i^{(t)}\|_2 \\
&\leq \beta_0 \|x^{(t+1)} - x^{(t)}\|_2 + \delta R^{(t)}/N \\
&\leq \beta_0 \big(\|x^{(t+1)} - x^*\|_2 + \|x^{(t)} - x^*\|_2\big) + \delta R^{(t)}/N \\
&\leq 2\beta_0 \mu^t W + \delta R^{(t)}/N \\
&= 2\beta \mu^t W/N + \delta \beta K \mu^t W/N \\
&= (2/K + \delta) K \beta \mu^t W/N \\
&= (\xi + \delta) K \beta \mu^t W/N \\
&\leq (\xi + \delta K) K \beta \mu^t W/N = K \beta \mu^{t+1} W/N = R^{(t+1)}/N.
\end{aligned}$$

*Global quantisation (Q3)*: To prove (Q3), we start by giving two auxiliary inequalities. First, we prove that $\|\nabla F(x^{(t+1)}) - r^{(t+1)}\|_2 \leq \delta R^{(t+1)}/2$:

$$
\begin{aligned}
\|\nabla F(x^{(t+1)}) - r^{(t+1)}\|_2 &= \|\sum_{i=1}^{N} \nabla f_i(x^{(t+1)}) - \sum_{i=1}^{N} q_i^{(t+1)}\|_2 \\
&\leq \sum_{i=1}^{N} \|\nabla f_i(x^{(t+1)}) - q_i^{(t+1)}\|_2 \\
&\leq N\delta R^{(t+1)}/(2N) = \delta R^{(t+1)}/2 \,.
\end{aligned}
$$

Next, we want to prove $\|r^{(t+1)} - q^{(t+1)}\|_2 \leq \delta R^{(t+1)}/2$. Again, it is sufficient to show $\|r^{(t+1)} - q^{(t)}\|_2 \leq (1 + \delta/2)R^{(t+1)}$, as the claim then follows from the definition of $q^{(t+1)}$ and Corollary 10. We have

$$
\begin{aligned}
\|r^{(t+1)} - q_i^{(t)}\|_2 &= \|r^{(t+1)} + \nabla F(x^{(t+1)}) - \nabla F(x^{(t+1)}) + \nabla F(x^{(t)}) - \nabla F(x^{(t)}) - q^{(t)}\|_2 \\
&\leq \|r^{(t+1)} - \nabla F(x^{(t+1)})\|_2 + \|\nabla F(x^{(t+1)}) - \nabla F(x^{(t)})\|_2 + \|\nabla F(x^{(t)}) - q^{(t)}\|_2 \\
&\leq \delta R^{(t+1)}/2 + \beta\|x^{(t+1)} - x^{(t)}\|_2 + \delta R^{(t)} \\
&\leq \delta R^{(t+1)}/2 + R^{(t+1)} = (1 + \delta/2)R^{(t+1)} \,,
\end{aligned}
$$

where the last inequality follows from the argument used in the proof of (Q2).

Finally, putting things together, we have

$$
\begin{aligned}
\|\nabla F(x^{(t+1)}) - q^{(t+1)}\|_2 &= \|\nabla F(x^{(t+1)}) - r^{(t+1)} + r^{(t+1)} - q^{(t+1)}\|_2 \\
&\leq \|\nabla F(x^{(t+1)}) - r^{(t+1)}\|_2 + \|r^{(t+1)} - q^{(t+1)}\|_2 \\
&\leq \delta R^{(t+1)}/2 + \delta R^{(t+1)}/2 = \delta R^{(t+1)} \,,
\end{aligned}
$$

completing the proof. $\qquad\square$

**Lemma 15.** *For any $\varepsilon > 0$ and $t \geq 2\kappa \log \frac{W}{\varepsilon}$, we have $\|x^{(t)} - x^*\| \leq \varepsilon$.*

*Proof.* By Lemma 14, we have $\|x^{(t)} - x^*\|_2 \leq \mu^t W = (1 - (1 - \mu))^t W \leq e^{-(1-\mu)t}W$. Assuming $t \geq \frac{1}{1-\mu} \log \frac{W}{\varepsilon}$. we have

$$
e^{-(1-\mu)t}W \leq e^{-(1-\mu)(1-\mu)^{-1}\log W/\varepsilon}W = e^{\log \varepsilon/W}W = \varepsilon W/W = \varepsilon \,.
$$

The claim follows by observing that $\frac{1}{1-\mu} = 2\kappa$ by definition. $\qquad\square$

**Communication cost.** Finally, we analyse the distributed implementation described at the beginning of this section, and analyse its total communication cost. Recall that we assume that the parameters $\alpha$ and $\beta$ are known to all nodes, so the parameters of the quantised gradient descent can be computed locally, and use $W = d^{1/2}$. Note that $W$ is the only parameter depending on the input domain, so the algorithm also applies for arbitrary convex domain $\mathbb{D} \subseteq \mathbb{R}^d$, setting $W$ to be the diameter of $\mathbb{D}$.

Since $\delta < 1$, we have by Lemma 10 that the each of the messages sent by the nodes has length at most $O(d \log \delta^{-1})$ bits. Assuming $\kappa \geq 2$, we have

$$
\log \delta^{-1} = \log \frac{2\kappa}{1 - \kappa^{-1}} \leq \log 8\kappa \,.
$$

Since the nodes send a total of $2N$ messages of $O(d \log \kappa)$ bits each, the total communication cost of a single round is $O(Nd \log \kappa)$ bits.

To get $\|F(x^{(T)}) - F(x^*)\|_2 \leq \varepsilon$, we need $\|x^{(T)} - x^*\| \leq (\varepsilon/\beta)^2$. By Lemma 15, selecting $T = O(2\kappa \log \frac{\beta W}{\varepsilon})$ is sufficient. Finally, using $W = O(d^{1/2})$, we have that the total communication cost of the optimisation is $O\big(Nd\kappa \log \kappa \log \frac{\beta d}{\varepsilon}\big)$. For the transmission of the local function values $f_i(x^T)$, there are at most $(\beta_0 d + 1)N/\varepsilon$ possible values, so each node needs to send $O(\log \beta d/\varepsilon)$ bits.

