# OpenReview forum: "Improved Communication Lower Bounds for Distributed Optimisation"
_ICLR.cc/2021/Conference — Reject_

### Official Review · AnonReviewer4 · 2020-10-25
**An interesting theoretical work with a lack of background and motivation**

**Rating:** 6
**Confidence:** 3

**Review:**

This paper studies the problem of optimizing a sum of $\beta$-strongly convex and $\alpha$-strongly smooth functions in the distributed point-to-point communication setting. It provides deterministic and randomized lower-bounds on the communication bit complexity and a quantized-based algorithm with asymptotic matching upper-bounds (for constant $\beta/\alpha$).

On the positive side, the paper contributes to a line of work which has significant practical relevance -- performing optimization over a set of functions distributed across many independent computing devices. It studies a specific task in this context from the point of minimizing the communication cost.


Regarding the model: I don't understand why the paper makes a setup about "message-passing model" just to say that their setup is equivalent to "coordinator model". Why not from the beginning say that it uses the "coordinator model"?

Theorem 4 seems to assume a (rather strict) regime where $d > O(N)$. That is, for $d \beta / (N^2 \varepsilon)$ to be a constant, and where $\beta = \beta_0 N$, we have $d > O(N \varepsilon / \beta_0)$. Is it justified to assume that the input dimension is at least as large as the number of devices? I would expect the situation to be the opposite.

This is a mainly theoretical work, but it still would be nice to motivate why the setup/task considered in this work (e.g., minimizing $\sum_i f_i(x)$ where each machine/device holds potentially different functions) is practically relevant. If it is mainly of theoretical relevance, then it would be nice to give reasons why this optimization task is an important one. This is not necessarily a critic on the relevance of the problem, but rather a comment that having more background would enable a reader to put the results into a broader perspective.

Regarding related work, what are other optimization tasks on a set of functions that would "make sense" to consider next? What are the ones that are important to understand, either theoretically or practically?

The appendix has a substantial amount of typos; not all of them are noted below.



Typos and suggestions:

Page 2: "... is somewhat subtle, so defer the details..." -> "... is somewhat subtle, so we defer the details..."

Page 2: "... which allows arbitrary centring of iterates..." -> "... which allows arbitrary centering of iterates..."

Page 6 (Definition of $D$, between Theorem 4 and Definition 5): Do you mean "$D = \Omega(...)$?" instead of "$\Theta(...)$"?

Page 7: "... we use it works in..." -> "... we use works in..."

Page 11, Line 3 of Lemma 6 proof: MEAN^{\varepsilon, \beta}_{d,N} should be MEAN^{\varepsilon, 4 \beta}_{d,N}

Page 11, Line 5 of Lemma 6 proof: Should "$F(x) = ... = 4 \beta \| x-x^* \| + C$" be "$F(x) = ... = 4 \beta \| x-x^* \|^2$"?

Page 11, Line 2 of Lemma 8 proof: From what I understand, we need zeta to be in the packing set S so that we can "infer Alice's input" (first paragraph after the chunk of equations on Page 12). Why is it enough to be uniformly random on {0,1}^D?

Page 11, last line of Lemma 8 proof: Should $\zeta$ be $\zeta_{1/2}$?

Page 12, first line of (2) in Lemma 8 proof: Would be nice to explicitly state that "node 1" is where the "WLOG i=1" is used.

Page 13, definition of $f_b(x)$: I had a bit of difficulty understanding the definition of $f_b(x)$ as it is. Maybe define a mapping function $g$ from $S$ to $[|S|]$, then use $g(s)$ when indexing the bit-string $\{0,1\}^{|S|}$?

Page 13, line before Theorem 9: In paper: "... the functions $f_T$ are well-defined..." Do you mean: "... the functions $f_b$ are well-defined..."?

Page 15, line 1 in proof of Q1: "... + x^*||_2" -> "... - x^*||_2"

Page 15, line 2 to 3 in proof of Q1: It seems that the proof from lines 2 to 3 assumes that $x^{(t+1)} = x^{(t)} - \gamma \nabla F(x^{(t)})$. While this is true for performing gradient descent directly, why is it still true after the quantization stuff?

Page 15 (last text line) and Page 16 (4th text line): There are references to "definition of $q^{(t+1)}$ and Lemma 10". There is no Lemma 10 but only Corollary 10, which refers to the prior quantization result from Alistarh et al.
What is this definition referring to and why does it suffice to only show what is shown in the proofs?

Page 16, first 2 lines in the first chunk of equations: should $f_i(x^{(t+1)})$ have $\nabla$'s in front of them?

Page 16, 4th text line: "$\ldots \le (1 + \delta/2) R^{(t)}$" -> "$\ldots \le (1 + \delta/2) R^{(t+1)}$"

Page 16, line 1 in second chunk of equations: "$... - q^{(t+1)}\|_2$" -> "$... - q^{(t)}\|_2$"

Page 16, line 2 in second chunk of equations: "$\le \|R^{(t+1)} ... - r^{(t+1)}\|_2$" -> "$\le \|r^{(t+1)} ... - q^{(t)}\|_2$"

---

> ### Author Response · Authors · 2020-11-18
> **Response to reviewer 4**
>
> The paper applies to any setting which trains convex or non-convex machine learning models in a distributed fashion. Thus, it is relevant to large-scale solvers for regression or generalized linear models, as well as distributed neural network training, though we note that the practical applications often further complications that are may  captured by the abstract model. We note that, in both these settings, the dimension $d$ is usually larger than the number of nodes N.
>
> More concretely, in the regression setting the dimension $d$ can vary in order of magnitude between 10 and 1000 (see e.g. [Stich 2018; arxiv 1805.09767] for an example), whereas neural networks regularly have millions of parameters. In both settings, the number of workers is usually small (2-16), but it can be scaled to the hundreds in extreme cases, e.g. [Goyal et al. 2017; arxiv 1706.02677].
>
> > Regarding the model: I don't understand why the paper makes a setup about "message-passing model" just to say that their setup is equivalent to "coordinator model". Why not from the beginning say that it uses the "coordinator model"?
>
> While for total communication cost, the models are equivalent, these are generally speaking two different models; typically, "message-passing" allows direct point-to-point messages between any pair of nodes, while the coordinator model allows only communication with the coordinator. If one considers complexity measures such as round complexity under limited bandwidth $B$, the models are not always equivalent.
>
> While this distinction does not matter for our work, we felt it was important to distinguish between the models. We will try and revise the discussion to make the relationship clearer.

---

> ### Author Response · Authors · 2020-11-18
> **Response to reviewer 4, technical comments**
>
> We thank the reviewer for thorough comments on the typos and technical details. We will incorporate the corrections and do another proof-reading pass. More detailed responses to select comments can be found below:
>
> > Page 6 (Definition of $D$, between Theorem 4 and Definition 5): Do you mean "$D = Ω(...)$" instead of "$Θ(...)$"?
>
> We can always select the set $S$ so that the equation holds with $Θ$ just by making $S$ smaller, though only the lower bound is needed for the rest of the proof.
>
> > Page 11, Line 2 of Lemma 8 proof: From what I understand, we need zeta to be in the packing set $S$ so that we can "infer Alice's input" (first paragraph after the chunk of equations on Page 12). Why is it enough to be uniformly random on $\{ 0,1 \}^D$?
>
> I am not sure if I understood this question correctly, but note that we have fixed a bijection between $\{ 0, 1 \}^D$ and the packing set $S$, so the distribution $\zeta$ can be thought of being the uniform distribution on $S$. For the proof to go through, we need $\zeta$ to be a hard distribution for the MEAN problem, and $\mu$ to be a product of independent $\zeta$ distributions.
>
> > Page 11, last line of Lemma 8 proof: Should $\zeta$ be $\zeta_{1/2}$?
>
> $\zeta$ as is defined to be $\zeta_{1/2}$ at the start of the proof; we will clarify this (see line 4 of the proof).
>
>
> > Page 13, definition of $f_b(x)$: I had a bit of difficulty understanding the definition of $f_b(x)$ as it is. Maybe define a mapping function $g$ from $S$ to $[|S|]$, then use $g(s)$ when indexing the bit-string $\{ 0,1 \}^{|S|}$?
>
> We will try to make the definition a bit clearer.
>
> > Page 15, line 2 to 3 in proof of Q1: It seems that the proof from lines 2 to 3 assumes that $x^{(t+1)}=x^{(t)}−\gamma \nabla F(x^{(t)})$. While this is true for performing gradient descent directly, why is it still true after the quantization stuff?
>
> There is yet another typo ($x^{(t+1)}$ -> $x^{(t)}$) on line 3. That is, the inequality applied to the second term is $|| x^{(t)}−\gamma \nabla F(x^{(t)}) - x^* ||_2 \le \sigma || x^{(t)} - x^* ||_2$, i.e. the standard gradient descent convergence bound. Note that this term is what an exact gradient descent would obtain without the quantisation error, and the quantisation error is isolated to the first term in the inequality; intuitively, the second term is distance bound for the exact step the algorithm tries to take, and the first term is the loss of convergence caused by the quantisation. The value $x^{(t)}−\gamma \nabla F(x^{(t)})$ is not actually computed by the algorithm.
>
> > Page 15 (last text line) and Page 16 (4th text line): There are references to "definition of $q^{(t+1)}$ and Lemma 10". There is no Lemma 10 but only Corollary 10, which refers to the prior quantization result from Alistarh et al. What is this definition referring to and why does it suffice to only show what is shown in the proofs?
>
> Lemma 10 should refer to Corollary 10. In the Local quantisation (Q2) step of the proof, we want to show that
>
> $|| \nabla f_i(x^{(t+1)}) - q^{(t+1)}_i ||_2 \le \delta R^{(t+1)}/(2N)$.
>
> As defined on page 15, $q^{t+1}_i$ is defined as
>
> $q^{(t+1)}_i = Q(\nabla f_i(x^{(t+1)}), q^{(t)}_i, R^{(t+1)}/N, \delta R^{(t+1)}/(2N))$,
>
> i.e it is $\nabla f_i(x^{(t+1)})$ quantised wrt. $q^{(t)}_i$. By the accuracy property of the quantisation scheme (Corollary 10(2)), $|| \nabla f_i(x^{(t+1)}) - q^{(t+1)}_i ||_2 \le \delta R^{(t+1)}/(2N)$ holds when $\nabla f_i(x^{(t+1)})$ and $q^{(t)}_i$ are sufficiently close to each other, i.e. $||\nabla f_i(x^{(t+1)}) - q^{(t)}_i||_2 \le R^{(t+1)}/N$ holds.
>
>
> > Page 16, first 2 lines in the first chunk of equations: should $f_i(x^{(t+1)})$ have $\nabla$'s in front of them?
>
> Yes.

---

### Official Review · AnonReviewer3 · 2020-10-29
**Theoretical results on communication of convex (and non-convex) optimization; may be of interest?**

**Rating:** 5
**Confidence:** 3

**Review:**

The submission studies the theoretical coordinator model communication complexity of multiple parties, each holding a strongly convex f_i, for the task of approximately minimizing the sum f_i.
The main result is a communication lower bound, which basically follows by picking the approximation parameter so restrictive that each party has to send the minimum of its f_i, to appropriate precision, in all dimensions.
*******************************************
Communication complexity is an important bottleneck for optimization. The theoretical developments in this paper are not particularly exciting and the writing is not great. This may still be of interest to ICLR community.
*********************************************

Abstract typo: “held by a one”

The comparison to the very recent paper of Vempala et al. (2020) is a little unfair IMHO. The authors say that the main differences are (i) that Vempala et al.’s problem has natural combinatorial encoding, but that is also true for the current paper; (ii) Vempala et al. use multiplicative vs additive approximation, but the approximation requirements in this paper are also very strong.

\sigma used as a standard deviation in the introduction, and something confusing in technical sections

Proof of Theorem 3: how do you get the factor of N in log|S|?

Proof of Theorem 3: “Since communication is only used for the simulation of Π”
Your reduction also needs communication to decide whether F(y)=0 or not.

ED not formally defined

---

> ### Author Response · Authors · 2020-11-18
> **Response to reviewer 3**
>
> Regarding the comparison to Vempala et al., we apologise if it seemed like we were being unfair to this paper, which we found extremely interesting, and was in fact one of the motivating factors behind our current study. We are revising the discussion of this paper in the revision.
>
> To give further context for our comparison, we wrote the comparison from a machine learning perspective, where the input functions are allowed to be essentially arbitrary real-valued functions, which do not necessarily have natural encodings. (More precisely, certain kids of circuits  could be used to represent restricted function classes, but we would still want to allow constant real terms in the circuit representation). In particular, the upper bound holds for arbitrary real functions as long as the model conforms to the setting described in Section 2. However, it is a fair point that the lower bounds are ultimately based on a discrete instance, though it uses a rather unnatural encoding of the input/output point sets.
>
> We also note that the additive approximation is necessary in the real-valued setting, as we cannot send arbitrary real numbers exactly over a binary channel. One can, roughly speaking, view the approximation factor $\varepsilon$ as a precision parameter of the output. We would also mention that additive approximation constraints appear to be more common in machine learning applications.
>
> Responses to minor comments:
>
> > \sigma used as a standard deviation in the introduction, and something confusing in technical sections
>
> Correct, we will revise.
>
> > Proof of Theorem 3: how do you get the factor of N in log|S|?
>
> This is an editing error, there should be no N in the expression. Will fix.
>
> > Proof of Theorem 3: “Since communication is only used for the simulation of Π” Your reduction also needs communication to decide whether F(y)=0 or not.
>
> This is an excellent point; we are implicitly making a (standard) assumption that the algorithms also output the (approximate) value of the target function $F(x)$. That is, upon the completion of the algorithm, the coordinator should output a point $z$ satisfying $\sum_{i = 1}^N f_i(z) \le \inf_{x \in [0,1]^d} \sum_{i = 1}^N f_i(x) + \varepsilon$, as well as an value $r$ satisfying
> $\sum_{i = 1}^N f_i(z) \le r \le \sum_{i = 1}^N f_i(z) + \varepsilon$.
> In practice, this means that the coordinator returns the "trained" machine learning model, and the corresponding value of the loss function, which is what one would expect.
>
> We apologise for not making this clear, and will make this explicit in the revision.
>
> > ED not formally defined
>
> ED is the expected distributional expected communication complexity, defined in Section 2 (page 5, line 3.) We will clarify.

---

### Official Review · AnonReviewer2 · 2020-11-02
**Degree of advance or surprise over prior work is not clear.**

**Rating:** 5
**Confidence:** 2

**Review:**

This paper studies the minimum number of bits that need to be communicated between N machines that jointly seek to minimize \sum_{i=1}^N f_i(x) where each function f_i is held by one of the machines, and the domain D of each f_i is a subset of R^d. The motivation for this is large optimization tasks that have to be solved often in machine-learning problems: the hope is to learn the limits to which the number of bits communicated in popular machine learning tasks can be optimized.

While the problem studied in the rest of the paper is cast purely in theoretical terms in the classical Message-Passing setting in distributed optimization, and therefore a conference with more theoretical bent seems more appropriate, given the many prior works that have appeared in machine learning conferences I am not too worried on this count.

As the main result, the paper shows that for quadratic optimization (i.e., even if all the f_i's are guaranteed to be quadratic functions),  to obtain an additive epsilon approximation (to the minimum of \sum_i f_i(x)) deterministically requires \Omega(Nd*log(beta*d/epsilon)) bits and the same for randomized approximation is \Omega(Nd*log(beta*d (N*epsilon))) bits. Here beta is the smoothness parameter of \sum_i f_i. The closest related earlier work showed that in the 2-node setting one requires \Omega(d*log(beta*d/epsilon)) bits. The generalization to N machines of this earlier result is a natural extension to study.

A couple of concerns:

1) Given the result for the two machine setting, what would one expect to be the lower bound in the N machine setting? Perhaps the proof maybe involved, but are there reasons to expect the lower bound to take any other form? If there are, they don't seem to be present in the paper.

2) Even if the result is not unexpected, proving it could well be complicated. So if there are clear technical innovations compared to prior work that would be a good plus. However the degree of technical advance is not quite clear to me --- I have not worked in this area to say for sure. The authors point out that the innovation is here is the connection built to communication complexity in the context of real-valued optimization-tasks, but I am not so sure if there is enough to clear the ICLR bar.

Overall this is a problem whose answer is good to know. The paper is written quite well, and is situated well in the landscape of related work. I am not sure if it clears the ICLR bar, perhaps accept if room.

---

> ### Author Response · Authors · 2020-11-18
> **Response to reviewer 2**
>
> We agree that the bounds we obtain are, in retrospect, what one would expect, especially given the two-machine lower bound of Tsitsiklis & Luo. However, as we discuss in the related work, there was no non-trivial lower bound for the N-machine setting, and existing proof techniques do not generalize, leaving this basic question open for some decades.
>
> Based on prior state of the art, one might conjecture for instance that the amount of information which needs to be sent by an individual machine could go down when the number of machines increases, as each local input function has less of an impact on the result; however, our results show that this is not the case. Indeed, one way to interpret the lower bounds is that the current quantised gradient descent methods are in fact the correct way to approach the problem. In general, this is what complexity lower bounds contribute to algorithm design: understanding the fundamental limits, allowing to distinguish between a case where we are facing hard limits of what is possible from the case where our algorithm development is stuck in a "local optimum."
>
> With the above discussion in mind, we view the main contributions of this paper to be (a) defining a rigorous model for studying the communication complexity of continuous optimisation, (b) the formal proofs of the lower bounds, and (c) connecting the machine learning setting to communication complexity; the technical details are a necessary secondary contribution. The current state of the art in communication complexity seems mostly sufficient to provide the technical tools we need for our results. However, as evidenced by the fact that these fundamental questions were not answered before, we argue that our conceptual contributions were exactly what was missing.

---

### Decision · Program_Chairs · 2021-01-10
**Final Decision**

**Decision:**

Reject

**Comment:**

This work presents an improved lower bound on the communciation complexity of distributed optimization in some settings. While reviewers agree that the paper is addressing a challenging and important question, all reviewers questioned the significance of the contributions of this work. In particular, two reviewers felt that the novelty of this work is limited. Unfortunately, the author response was unable to adequately address these concerns.